# Improved Friction and Wear Properties of Al6061-Matrix Composites Reinforced by Cu-Ni Double-Layer-Coated Carbon Fibers

**Shuhan Dong [1], Huiyong Yuan [2], Xiaochao Cheng [1], Xue Zhao [1],\*, Mingxu Yang [1,3],\*, Yongzhe Fan [1] and Xiaoming Cao [1]**

[1] School of Materials Science and Engineering, Hebei University of Technology, 29 Guangrong Road, Tianjin 300132, China; 18812039257@163.com (S.D.); m15822772448@163.com (X.C.); fyz@hebut.edu.cn (Y.F.); gd_sam@galvanize.com.cn (X.C.)

[2] Beijing Sunwise Intelligent Technology Co. Ltd., 16 Nansan Street, Zhongguancun, Beijing 100080, China; yuanhuiyong@sunwiserobot.com

[3] Beijing Jitai Cold-Forging Technology Co., Ltd., 9 Zhongguancun South Avenue, Beijing 100081, China

\* Correspondence: zhaoxue@hebut.edu.cn (X.Z.); yangmingxu@hebut.edu.cn (M.Y.)

**Abstract:** The friction and wear properties of an Al6061 alloy reinforced with carbon fibers (CF) modified with Cu-Ni bimetallic layers were researched. Cu-Ni double layers were applied to the CF by electroless plating and Al6061-matrix composites were prepared by powder metallurgy technology. The metal-CF/Al interfaces and post-dry-wear-testing wear loss weights, friction coefficients, worn surfaces, and wear debris were characterized. After T6 heat treatment, the interfacial bonding mechanism of Cu-Ni-CF changed from mechanical bonding to diffusion bonding and showed improved interfacial bonding strength because the Cu transition layer reduced the fiber damage caused by Ni diffusion. The metal–CF interfacial bonding strongly influenced the composite's tribological properties. Compared to the Ni-CF/Al and Cu-CF/Al composites, the Cu-Ni-CF/Al composite showed the highest hardness, the lowest friction coefficient and wear rate, and the best load-carrying capacity. The wear mechanisms of Cu-Ni-CF/Al composite are mainly slight abrasive wear and adhesive wear.

**Keywords:** friction and wear properties; Al6061-matrix composites; Cu-Ni bimetallic layers; carbon fibers; diffusion bonding

---

## 1. Introduction

Studies in aerospace and automotive fields have shown that the materials used for components like engines should possess good mechanical and tribological properties [1]. The application of metal matrix composites (MMCs) in dry friction environments requires good wear resistance [2]. Al alloy composites have high strength, high electrical and thermal conductivity, and good fatigue resistance [3]. However, their poor wear resistance limits their applicability.

Carbon fibers (CFs) are widely used in Al alloy-based MMCs because they couple high moduli and strengths with lubrication and abrasion resistance [4,5]. The self-lubricating properties and abrasion resistance of short carbon fibers (SCFs)-reinforced Al6061 alloy composites (SCFs/Al) are excellent [6,7]. However, the interfacial compatibility of SCFs and Al is poor. Interfacial reactions generating brittle $Al_4C_3$ occur easily, thus worsening the composites' mechanical properties [8,9]. The problem has been addressed by plating SCFs with Cu or Ni [10,11]. Such metal coatings improved the wetting behavior of SCFs with Al, thus improving the distribution uniformity of SCF reinforcements, reducing interfacial reactions, and improving composite hardness [12,13]. Metal coatings on SCFs positively affect the

mechanical and tribological properties of the composites [14,15]. The beneficial effects of Cu or Ni metal coatings on the wear behaviors of SCF-reinforced Al6061 alloy have been reported [16]. Ureña et al. [17] reported that Cu or Ni coatings were advantageous to wet the CF in the process before dissolving in the Al matrix to form intermetallic compounds, thus improving the hardness and wear resistance. They also found that the friction coefficient of Cu-SCF/Al was lower than that of Ni-SCF/Al. Xia et al. [6] found that Cu-SCF can improve the abrasion resistance of aluminum alloy.

However, present research only focuses on enhancing Al matrices by modified SCFs without heat treatment. In such composites, the fiber and metal coating are mechanically bonded with weak interfacial adhesion, so the strengthening effect of SCFs modified by a single metal layer (Cu or Ni) on the Al6061 alloy is not optimal. However, heat treatment of the coated SCFs induces Ni diffusion to the fibers, which leads to graphitization of the fiber structure [18], fiber damage, and weakened enhancement provided by the fibers to the composite material [19]. Intermetallic compounds ($Al_3Ni$) at the interface act as crack sources that deteriorate the bearing capacity and wear resistance of the composites [20]. Large pores form between the SCFs and Cu coating. The interfacial combination mode between the Cu coating and SCFs is low-strength mechanical bonding [17,21]. The weak interfacial bonding prevents the optimization of mechanical properties and fiber debonding occurs easily in friction and wear testing [22]. Therefore, the abrasion resistance of Al6061 alloy composites reinforced by SCFs modified with a single metal layer is poor.

The above results showed that using SCFs improved the specific modulus, strength, and load-transfer capability of composite structural components [23]. This indicated that SCFs, as reinforcements, effectively improved the wear resistance of Al-matrix composites. However, no researchers have studied the effects of Cu-Ni-modified SCFs, or the effects of SCF-coating interfacial characteristics on the tribological properties of SCF/Al composites.

In this study, SCFs were modified by electroless plating with Cu, Ni, or Cu-Ni and SCF/Al composites were prepared by powder metallurgy technology. The friction and wear behaviors of the composites reinforced by different modified SCFs were discussed systematically. The results demonstrated that interfacial bonding between the Cu-Ni double layers and SCF occurred by diffusion bonding, which is higher in strength than mechanical bonding. In addition, the Cu-Ni double layers formed a Cu-Ni solid solution after heat treatment. The solid solution was uniformly distributed in the SCF/Al interface and further improved the composite performance.

## 2. Experimental Details

### 2.1. Materials

Al6061 was used as the matrix alloy, and its composition is given in Table 1. The average particle size of the Al6061 alloy powder is 30 μm. The short carbon fibers (SCFs T300) were supplied by Japan Toray Co. Ltd. (Tokyo, Japan). The properties of the polyacrylonitrile-based SCFs are given in Table 2.

**Table 1.** Chemical Composition of Al6061 Alloy.

| Element | Si | Fe | Cu | Mg | Mn | Zn | Cr | Sn | Ti | Ni | Pb | Al |
|---|---|---|---|---|---|---|---|---|---|---|---|---|
| Percentage (wt.%) | 0.5 | 0.7 | 0.2 | 0.8 | 0.15 | 0.01 | 0.18 | 0.001 | 0.02 | <0.05 | 0.02 | Balance |

**Table 2.** Properties of Polyacrylonitrile (PAN)-Based SCFs.

| C/(%) | $\sigma_b$/(MPa) | E/(GPa) | D/(μm) | P/(g·cm$^{-3}$) |
|---|---|---|---|---|
| 98.5 | 3280.5 | 201.1 | 6.9 | 1.76 |

## 2.2. Electroless Cu, Ni, and Cu-Ni Coating of SCFs

The SCFs were coated with Cu or Ni single layers and Cu-Ni double layers by electroless plating. The treatment process for electroless plating includes the steps of degumming, deoiling, dispersion, coarsening, neutralization, sensitization, activation, reduction, and electroplating. Tables 3–5 show the experimental conditions and composition of the Ni, Cu, and Cu-Ni electroless plating solutions, respectively. Then, the modified SCFs were heat-treated in a ZT-40-20y vacuum sintering furnace at 750 °C and $10^{-2}$ Pa with a holding time of 50 min, followed by furnace cooling.

**Table 3.** The Solution Composition and Experimental Conditions of Electroless Nickel Plating.

| Stage and Conditions | Concentration of Chemicals |
|---|---|
| Metallization<br>pH (8–9)<br>Temperature (65–70 °C)<br>Time (5 min) | 25 g/L $NiSO_4 \times 2H_2O$<br>30 g/L $NH_4Cl$<br>25 g/L $NaH_2PO_2 \times H_2O$<br>25 g/L $Na_3C_6H_5O_7 \times 2H_2O$<br>1 mg/L $CH_4N_2S$<br>$NH_3 \cdot H_2O$ to control the pH |

**Table 4.** The Solution Composition and Experimental Conditions of Electroless Copper Plating.

| Stage and Conditions | Concentration of Chemicals |
|---|---|
| Metallization<br>pH (12–13)<br>Temperature (55–60 °C)<br>Time (5 min) | 25 g/L $CuSO_4 \times 5H2O$<br>20 g/L EDTA<br>20 g/L NaOH<br>15 mg/L $K_4Fe(CN)_6 \times 3H_2O$<br>20 mg/L $C_{10}H_8N_2$<br>15 mL/L HCHO |

**Table 5.** The Solution Composition and Experimental Conditions of Electroless Copper–Nickel Double Plating.

| Stage and Conditions | Concentration of Chemicals |
|---|---|
| Metallization<br>pH (9)<br>Temperature (65–70 °C)<br>Time (10 min) | 10 g/L $CuSO_4 \times 5H_2O$<br>2–3 g/L $NiSO_4 \times 2H_2O$<br>30 g/L $H_3BO_3$<br>10 mg/L $C_{10}H_8N_2$<br>50 mg/L $C_6H_5SO_2Na$<br>25 g/L $Na_3C_6H_5O_7 \times 2H_2O$<br>30 g/L $NaH_2PO_2 \times H_2O$<br>6.5 mg/L $C_{14}H_{14}N_3SO_3Na$ |

## 2.3. Composite Manufacturing

The composites were prepared by the following powder metallurgy processing. In the molding process, the mixture is prepressed at 100 MPa for 1 min and then compressed at 500 MPa for 5 min. After demolding, the formed blank is obtained. The blank is sintered in an evacuated tube furnace. The billet is placed in a vacuum tube furnace (argon atmosphere) for vacuum sintering (pre-firing at 400 °C for 1 h, sintering at 520 °C for 2 h) and cooled with the furnace. After sintering, the composite material is subjected to T6 heat treatment (solid solution treatment for 2 h at 500 °C → hydrocooling at 50 °C → artificial aging treatment for 10 h at 180 °C).

## 2.4. Wear Tests

The instrument used in the friction and wear experiment is an SFT-2M pin–plate friction and wear tester (China Lanzhou Zhongke Kaihua Technology Development Co., LTD. Lanzhou, China).

The AISI 52100-type bearing steel (outer diameter 3 mm) is used as the rotating ring. The size of the sample is Ø 20 mm × 5 mm. The specific experimental environment is as follows: the load is 10 N, the rotational speed is 400 r/min, the temperature is 293 ± 3 K, and the friction time is 20 min. Before and after the experiment, the sample is ultrasonically cleaned with alcohol and weighed with an electronic balance precise to 0.1 mg.

### 2.5. Composite Characterization

The coating components of the modified SCFs were characterized by X-ray diffraction (XRD; Rigaku Dmax 2500Pc, Tokyo, Japan) using monochromatic Cu Kα1 radiation of λ = 1.5418 Å, scanning speed of 6°/min, and a scanning power of 2 kW. The worn surface morphologies were examined by scanning electron microscopy (SEM, Quanta 450 FEG, Hillsboro, OR, USA, operating at 5 kV) and energy-dispersive X-ray spectrometry (EDS; Quanta 450 FEG, Hillsboro, OR, USA, operating at 10 kV).

The hardness of the material reflects the performance of the sintered sample. This experiment used a Shimadu HMV-2t Vickers (Tokyo, Japan) microhardness tester ($HV_{0.1}$ = 100 MPa, 10 s) to test the hardness of the composite; ten points were selected for each sample to be tested and averaged.

## 3. Results and Discussion

### 3.1. Microstructure of Modified SCFs

Figure 1 shows the interface microstructures of the modified SCFs. Each SCF is completely coated with metal with no shedding, indicating that the mechanical bonding between the fiber and coating is strong. There are changes in the element composition at the interface, as shown in the Figure 1a,c, along the line sweep direction, the C content is obviously increased while the Ni or Cu content is obviously decreased. The place where the element content changes is the interface between the SCF and the coating.

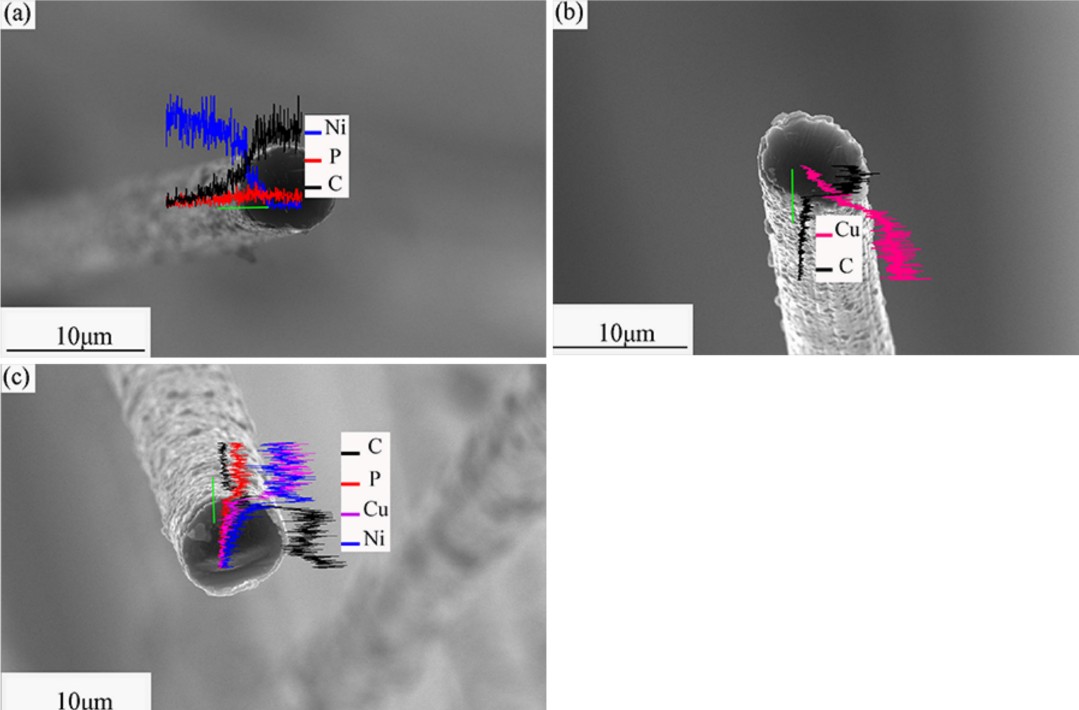

**Figure 1.** Scanning electron microscopy (SEM) images of the interface microstructures of the modified short carbon fibers (SCFs): (**a**) Ni-Cf; (**b**) Cu-Cf; and (**c**) Cu-Ni-Cf.

### 3.2. Heat Treatment of Modified SCFs.

Figure 2 shows the surface microstructures of the modified SCFs and EDS images of the modified SCF surfaces. Before heat treatment, the metal layers are smooth, uniform, and continuous. After heat treatment, the surface of the Ni coating shows nodules. The unheated Ni coating is in a metastable thermodynamic state. During the heat treatment, the mutual diffusion of Ni and C causes the coating to shrink and become rough. However, heat treatment does not significantly change the surface of the Cu coating because the solid solubility of C in Cu is very low and no chemical reaction occurs between C and Cu. On the surface of the heat-treated Cu-Ni coating, nodules are observed. Because mutual C-Ni diffusion occurs, an infinite solid solution reaction between Cu and Ni occurs at 750 °C to form a Cu-Ni binary alloy [24]. The microstructure of the Cu-Ni-SCF is thus greatly changed.

After heat treatment, the C peak becomes sharper for the Ni-SCF. During heat treatment, mutual C-Ni diffusion occurs, thus incorporating C into the Ni coating and significantly changing the Ni-SCF surface morphology. The Cu peak is basically unchanged after heat treatment, with no new components appearing, showing that C and Cu do not experience mutual diffusion. After heat treatment, the C peak becomes sharper and a Cu peak appears for the Cu-Ni-SCFs. This confirms the previous analysis: Ni and Cu react to generate the Cu-Ni alloy and C disperses into the Cu-Ni coating.

Figure 3 shows the interface microstructures of the modified SCFs. Before heat treatment, gaps are present between the SCFs and the metal coatings. The fibers and coatings are mechanically bonded. After heat treatment, the Ni-SCF interfacial gap disappears. Mutual diffusion between Ni and C changes the interfacial bonding mechanism from mechanical to diffusion-based. After heat treatment, the Cu coating remains separated from the SCF, indicating only mechanical bonding. In the Cu-Ni-SCF after heat treatment, no gap appears and the interfacial bonding mechanism is diffusion-based, because Ni diffusion reduces interfacial tension between Cu and the SCF. Meanwhile, the Cu-Ni solid solution somewhat impedes the diffusion of Ni, thus reducing the fiber damage caused by Ni diffusion.

The XRD results of the modified SCFs are shown in Figure 4. After heat treatment, no carbide phases are present, indicating that the SCF does not react with the metal coating. Cu, Ni, and a Cu-Ni alloy exist on the surfaces of the Cu-SCF, Ni-SCF, and Cu-Ni-SCF, respectively. The C peak in the Cu-SCF pattern is the lowest in intensity, indicating that mutual diffusion does not occur between C and Cu. The interface mode is mechanical bonding. The C peak in the Ni-SCF pattern is the highest in intensity, showing that the diffusion of Ni into C causes fiber graphitization. The C peak in the Cu-Ni-SCF pattern is lower than Ni-SCF in intensity. This means that the Cu coating hinders the diffusion of Ni into C and reduces the extent of graphitization. It also indicates that diffusion of Ni to C continues. The interface is diffusionally bonded.

### 3.3. Composite Microstructures

The microstructures of the composites are shown in Figure 5. The interface between the Al6061 matrix and the SCF is smooth and continuous with no defects, indicating a well-integrated interface. There is damage at the interface of the Ni-SCF/Al composite because the diffusion of the Ni into the SCF during heat treatment destroys the fiber structure. The micro-voids at the interface of the Cu-SCF/Al composite are caused by the mechanical combination of the Cu layer and SCF. In the Cu-Ni-SCF/Al composite, as an intermediate transition layer, the Cu layer not only prevents the diffusion of Ni into the SCF and reduces the degree of fiber damage, but also forms a densely diffusion-bonded interface.

The white areas in the Cu-SCF/Al and Ni-SCF/Al composites are $CuAl_2$ and $Al_3Ni$, respectively. The white area in the Cu-Ni-SCF/Al composite comprises $Al_3Ni$, $CuAl_2$, and CuNi. Figure 6 shows the microstructures of SCFs/Al composites with different SCF contents. When the fiber content is less than 6 vol%, the fibers are uniformly dispersed in the matrix, the junction between fibers and matrix is excellent, and no obvious defects appear. However, when the fiber content is 8 vol%, fiber dispersion is not uniform in the mixing process, agglomeration occurs in the matrix, and small holes appear at the interface. Figure 7 shows the details of some typical defects that occur when the fiber content is 8 vol%, there are holes and defects at the joint cross section of fiber and matrix.

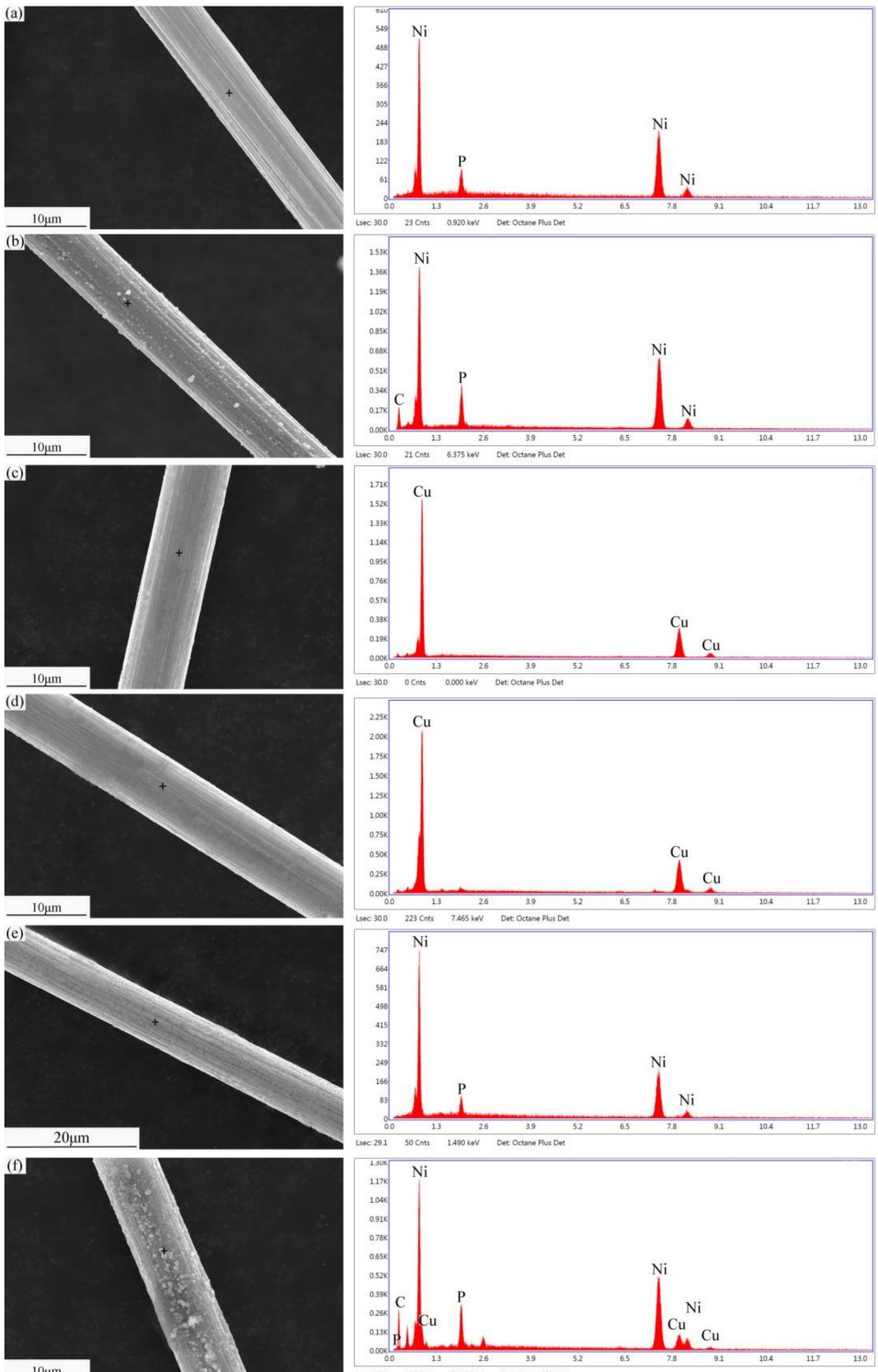

**Figure 2.** SEM images of surface microstructures of the modified SCFs and energy-dispersive X-ray spectrometry (EDS) images of the modified SCF surfaces: (**a**,**b**) Ni-Cf; (**c**,**d**) Cu-Cf; (**e**,**f**) Cu-Ni-Cf; (**a**,**c**,**e**) before heat treatment; and (**b**,**d**,**f**) after heat treatment.

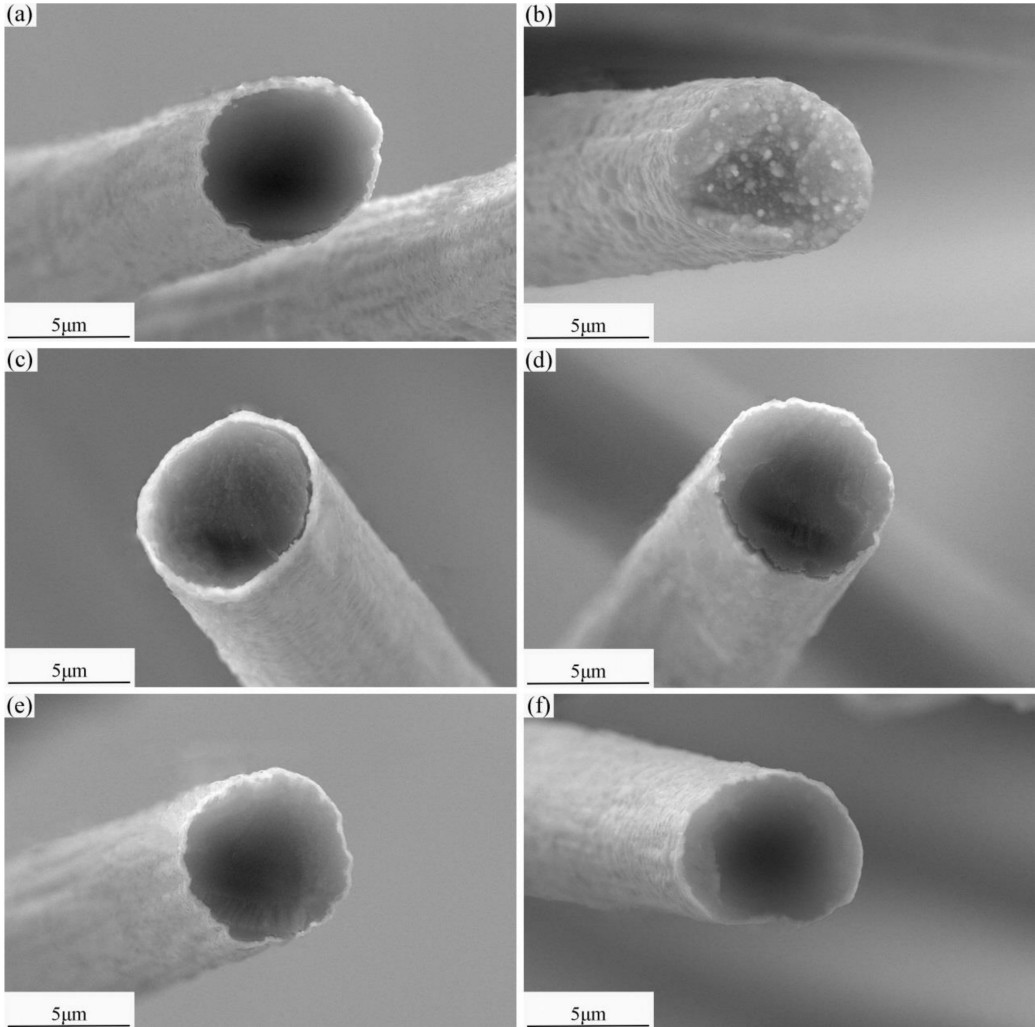

**Figure 3.** SEM images of the interface microstructures of the modified SCFs: (**a**,**b**) Ni-Cf; (**c**,**d**) Cu-Cf; (**e**,**f**) Cu-Ni-Cf; (**a**,**c**,**e**) before heat treatment; and (**b**,**d**,**f**) after heat treatment.

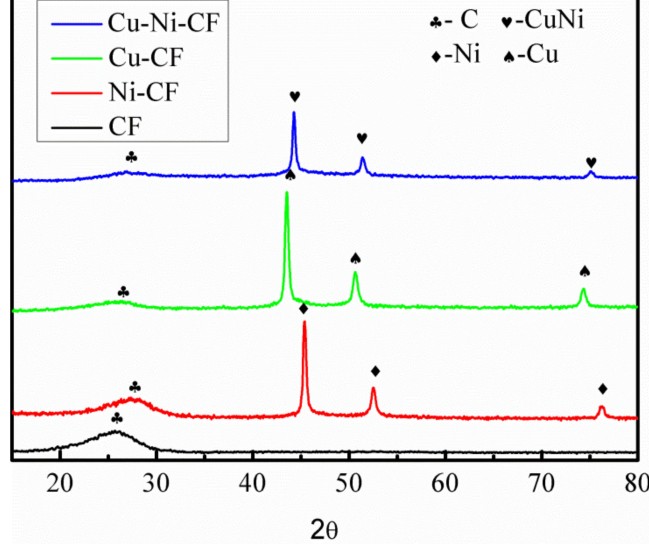

**Figure 4.** X-ray diffraction (XRD) patterns of carbon fiber with different coatings after heat treatment.

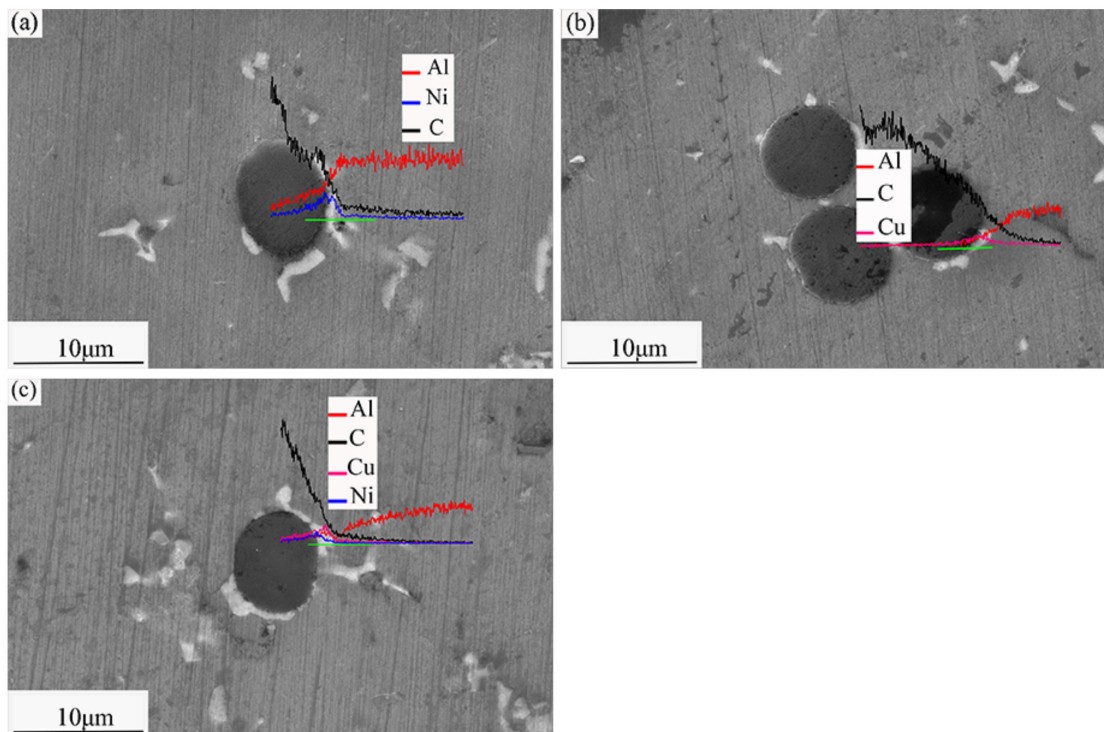

**Figure 5.** SEM images of microstructures of the composites: (**a**) Ni-Cf/Al; (**b**) Cu-Cf/Al; and (**c**) Cu-Ni-Cf/Al.

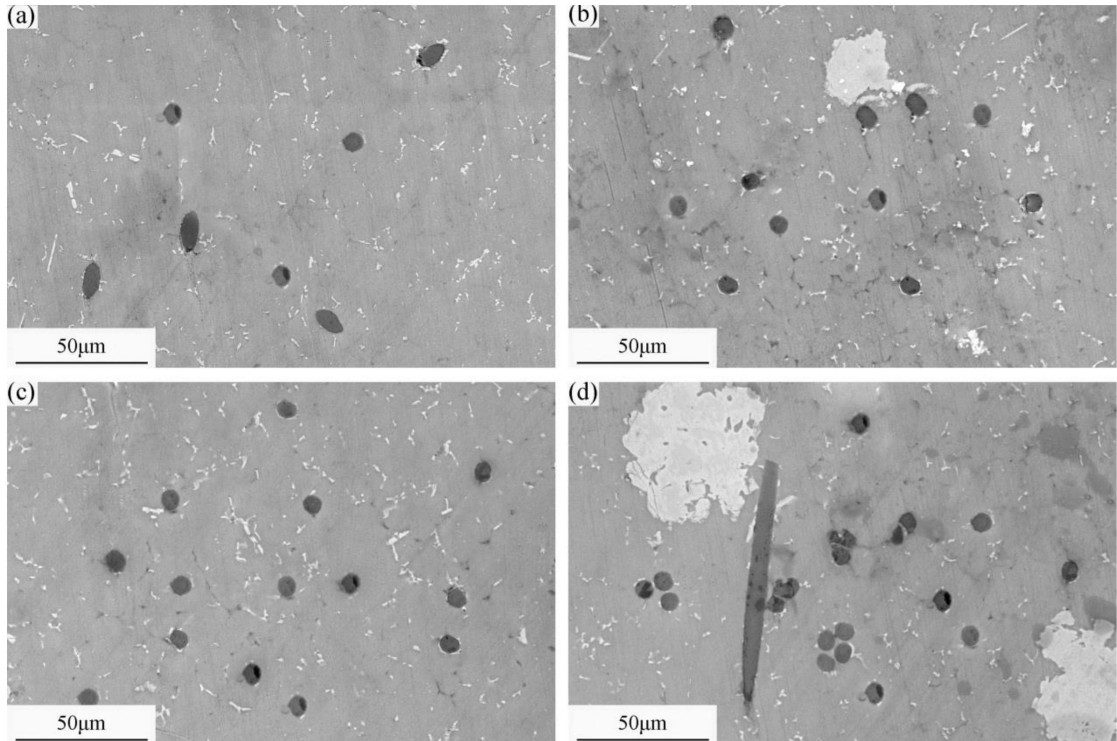

**Figure 6.** SEM images of the microstructures of SCFs/Al composites with different SCF contents: (**a**) 2%; (**b**) 4%; (**c**) 6%; and (**d**) 8%.

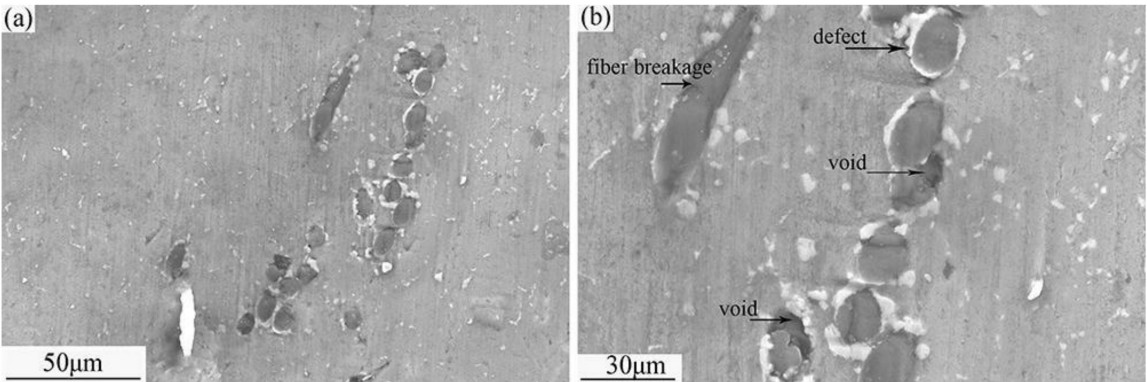

**Figure 7.** SEM microstructure of fiber agglomeration and defects in the aluminum matrix: (**a**) 2000×; and (**b**) 5000×. The fiber content is 8%.

### 3.4. Composite Hardness

Figure 8 shows the hardness of the different composites with different volume fractions of SCFs. The hardness of the coated SCF/Al composites obviously increased with increasing concentrations of coated SCFs. When the coated SCFs content is 6 vol%, the hardness values of the coated SCF/Al composites are maximized. The reasons are as follows: (1) The fibers are uniformly dispersed in the matrix and act as dispersive reinforcement. The coating prevents the formation of harmful phases at the interface between C and Al, and Cu and Ni diffuse into the matrix to form hard intermetallic compounds ($CuAl_2$ or $Al_3Ni$) to improve the overall hardness of the material. (2) Under the thermal action of the sintering process, because the thermal expansion coefficient of the SCF is greatly different from that of Al6061, the deformation degrees of the SCFs and Al6061 differ, which causes large geometric dislocations that hinder deformation. Therefore, the hardness of the composite material is increased with the increase of SCF content. The hardness values of the Ni-SCF/Al, Cu-SCF/Al, and Cu-Ni-SCF/Al composites are enhanced by 15.3, 18.5, and 23.3%, respectively, compared to that of the Al6061 alloy. For SCF contents surpassing 6 vol%, the composite hardness begins to decrease because increasing SCF aggregation produces many defects.

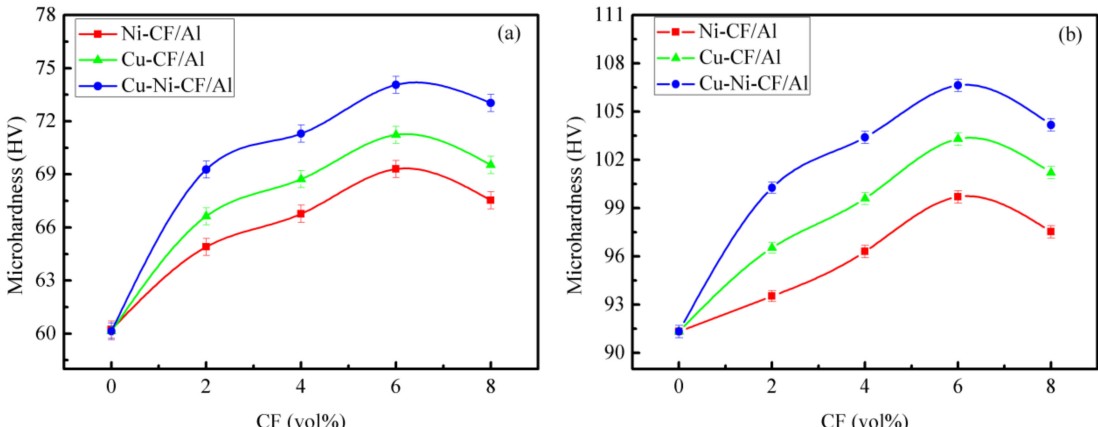

**Figure 8.** The hardness of different composites before and after heat treatment: (**a**) before heat treatment; and (**b**) after heat treatment.

The Cu-Ni-SCF/Al composite is the hardest because the Cu and Ni are spread on the SCF surface to form a Cu-Ni alloy after heat treatment. The Cu-Ni solid solution strengthens and thereby increases the hardness of the composite. After the Ni-SCF heat treatment, Ni diffusion causes fiber damage and weakens the effect of the fibers on the composites, so the Ni-SCF/Al composite is only harder than the

unreinforced Al6061 alloy. The Cu coating does not damage the fibers and Cu has a good strength retention rate; therefore, the Cu-SCF/Al is harder than the Ni-SCF/Al composite.

The composite hardness is improved after the T6 heat treatment for the following reasons:

1.  After the solid solution treatment, the high-concentration solid solution is transformed via the rapid cooling process to a supersaturated solid solution, yielding the solid solution strengthening effect. After the aging treatment, the phases of β and α form a common lattice strain region in the matrix, which impedes dislocation movement and improves the deformation resistance of the material.

2.  Because of the difference in the thermal expansion coefficient between the SCF and the matrix, the high-density dislocations generated at the interface are favorable for the heterogeneous nucleation of the precipitated strengthening phase, and act as short-circuit diffusion channels to improve the diffusion speed of solute atoms and promote the nucleation and growth of the precipitated phase. The interaction of these two aspects is shown as the acceleration of precipitation strengthening kinetics on the microscale level and the acceleration of hardening on the macroscale level.

The evolution of the friction coefficients of the composites is shown in Figure 9a. The friction coefficient of the unreinforced Al6061 alloy is the highest, because of the contact friction between Al6061 alloy and the metallic grinding ball in wear testing. During the wear test, the temperature of the Al6061 alloy surface is increased, which softens the material. As is shown in Figure 10a, the Al6061 alloy shows a worn surface with many grooves because the low strength of the alloy causes quality deterioration and produces adhesive wear during the wear process. This adhesive wear indicates that the friction coefficient of the Al6061 alloy is the highest.

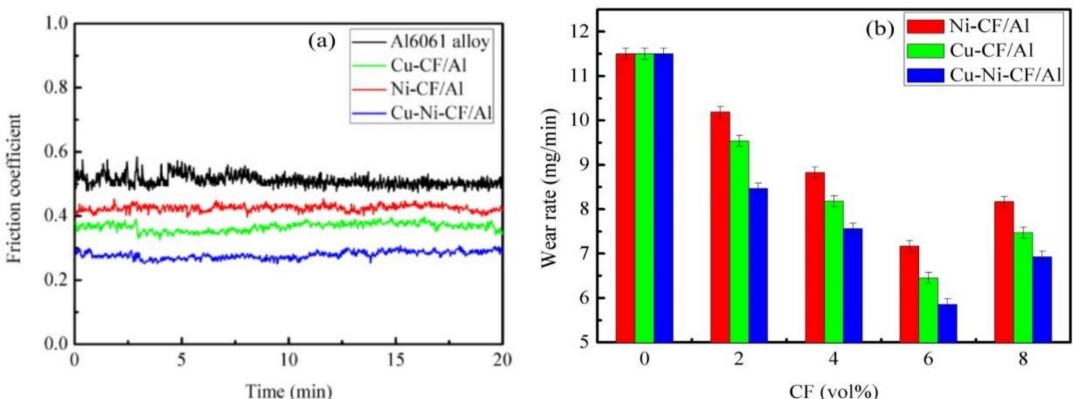

**Figure 9.** (**a**) The friction coefficient of Cf/Al6061 alloy composites with different coatings. (**b**) The diagram of wear rate and fiber content of different composite materials.

The composites have lower friction coefficients than the Al6061 alloy, because the SCFs comprise microcrystallites of graphite that act as lubricants [25]. As is shown in the energy spectra of Figure 12b–d, C is found on the worn surfaces of the composites, indicating that graphitic SCFs are ground to form a carbon film that spreads on the worn surface. During the wear test, the SCF/Al composite surfaces experience slight plastic deformation; the SCFs are ground and spread over the worn surface. These graphite films block direct-contact friction between the two metal surfaces and reduce heat production. The SCFs absorb energy, reduce the increases in surface temperature, and prevent composite softening. Furthermore, the metal coated on the SCFs is also sheared by the AISI 52100 steel grinding ball, because the metal plating is soft. In addition, metallic oxides may form during wear testing. All these factors prevent immediate contact friction between the two grinding surfaces, thereby reducing the friction coefficient of the composites.

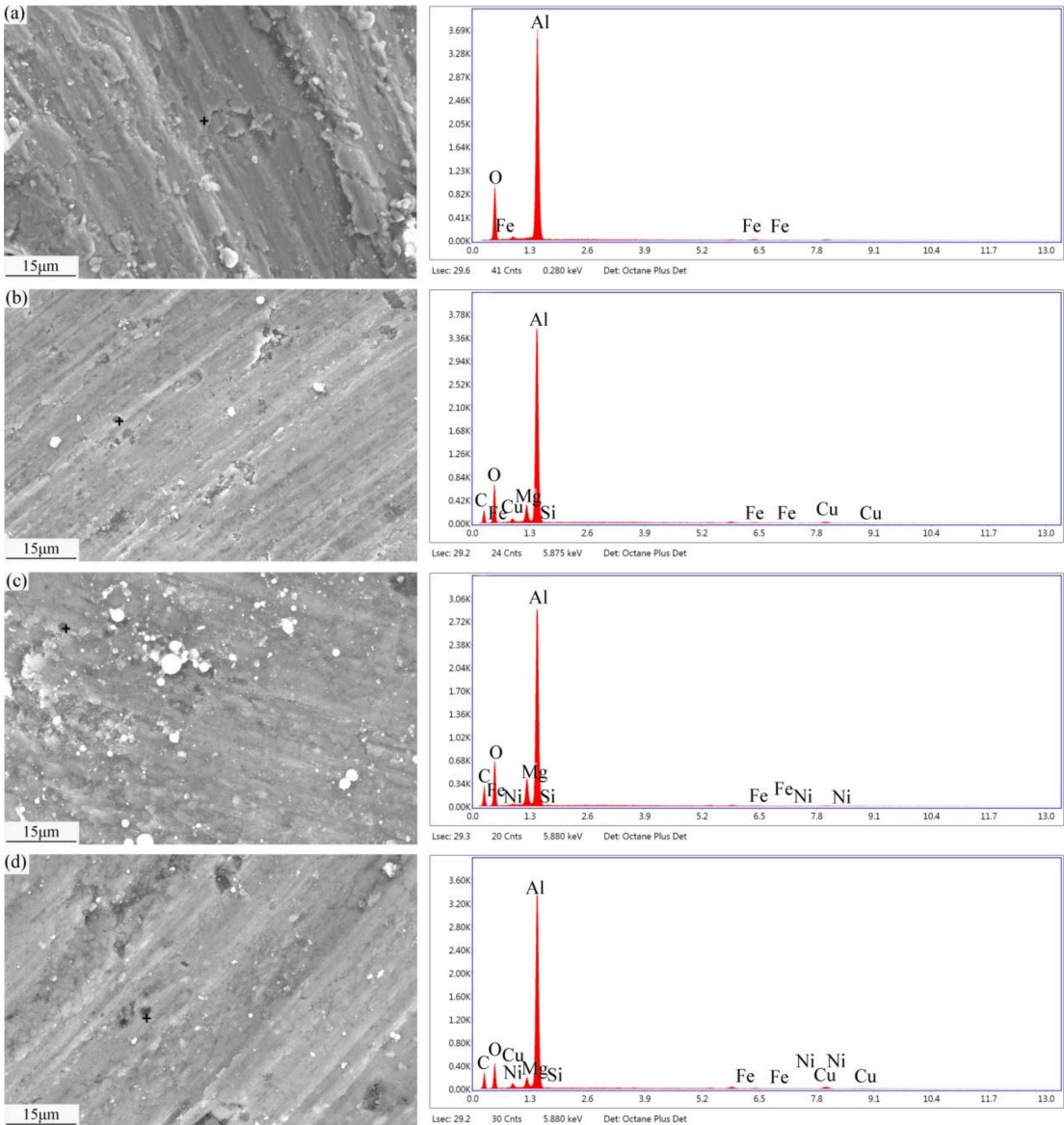

**Figure 10.** SEM images and EDS patterns of the worn surfaces of composites: (**a**) Al6061 alloy; (**b**) Cu-Cf/Al; (**c**) Ni-Cf/Al; and (**d**) Cu-Ni-Cf/Al.

Figure 9a shows that the composites friction coefficients increase in the following order: Cu-Ni-SCF/Al < Cu-SCF/Al < Ni-SCF/Al < Al6061 alloy.

The reasons can be summarized as follows:

1. Influence of the composite hardness: Improved composite hardness can enhance wear resistance and prevent the serious wear caused by plastic deformation. The hardness of the composites proceeds in the order Cu-Ni-SCF/Al > Cu-SCF/Al > Ni-SCF/Al > Al6061 alloy. Therefore, the abrasive resistance of the Cu-Ni-SCF/Al composite is the highest and its friction coefficient is the lowest.

2. Good adhesion between the coated SCFs and the Al6061 alloy matrix: The composites show higher interfacial bonding strengths, load-carrying capacities, and wear resistances [26]. The interfacial bonding strengths of the composites increase in the order Ni-SCF < Cu-SCF/Al < Cu-Ni-SCF/Al. After heat treatment, Ni diffusion in the Ni-SCF causes fiber damage that destroys the SCF structures and weakens the strengthening effect of CF on composites. The interfaces in the Cu-SCF/Al composite are weakly mechanically bonded. However, in Cu-Ni-SCF, Cu as an

intermediate transition layer hinders Ni diffusion and forms a diffusion-bonded interface. The interfacial combination between the matrix and reinforcement is better and thus the wear resistance of the MMC is better [27,28]. With a higher interfacial bonding strength, the SCFs are more difficult to remove from the Al matrix and the resistance to shear friction, deformation, and fracture are improved.

3. Effect of mechanical mixing friction layers: The significant improvement in the abrasion resistance of the composite is due to the formation of mechanical mixing friction layers (MMLs) comprising fine mixtures of hard intermetallic compounds, metallic oxides, and carbon fragments. Various metal oxides and SCF fragments are mixed in MMLs on the composite surfaces, thus enhancing the composite wear resistance [29]. During the wear tests, the materials are softened as the surface temperatures are increased. However, C and Cu have good thermal conductivity and reduce the damage caused by the softening effect. Therefore, the wear resistances of the Cu-Ni-SCF/Al and Cu-SCF/Al composites are better than that of the Ni-SCF/Al composite.

From the above, the hardness and interfacial bonding strength of the Cu-Ni-SCF/Al composite are the highest. Further, the C and Cu in the MMLs increase the thermal conductivity and reduce the damage caused by softening. Therefore, the wear resistance of the Cu-Ni-SCF/Al composite is the best as its friction coefficient is the lowest. Figure 9b shows that the wear rate of the Cu-Ni-SCF/Al composite is the lowest because the composite shows the highest hardness and the best wear resistance. The wear rate of a composite is closely related to the hardness, and wear is gradually decreased with increases in fiber content. However, for excessively high fiber contents, defects cause decreases in hardness and deteriorations in wear resistance, so the wear rate increases.

### 3.5. Wear Mechanisms

The SEM images and EDS patterns of the worn surfaces of composites are shown in Figure 10. The Al6061 alloy shows a worn surface containing many grooves because the low strength of the alloy causes quality deterioration and produces adhesive wear during the wear process. Under wear testing, adhesion and furrows create resistance to wear and tear that impedes the motion of the friction pair. The constant friction increases the surface temperature of the material and thus induces softening. Plastic deformation occurs and accumulates, generating holes and cracks. The cracks reach a certain critical depth because the stress is parallel to the surface. Wear debris is produced. During the wear tests, the energy is converted into both strain that induces surface deformation and heat that induces wear surface heating and oxidation. The appearance of oxygen indicates oxidative wear. For an external force exceeding the shear strength of the material, the surface is spalled with severe wear. The wear mechanisms are delamination and severe abrasion.

The worn surfaces of the SCF/Al composites are smoother and show shallower polishing scratches than those observed on the Al6061 alloy. The worn surfaces of the composites show tearing and slight delamination. Additionally, shallower plastic deformation zones exist near the polishing grooves. The reasons are as follows:

1. Under the action of external force, SCFs are extruded and ground into fine particles that form a lubricating carbon film on the worn surface. The pinning effect of the SCFs restricts matrix deformation and improves the deformation resistance of the composite, thus reducing crack generation.

2. The MML formed on the worn surface includes hard intermetallic compounds, such as $Al_3Ni$ and $CuAl_2$, and metal oxides, such as $CuO$ and $Al_2O_3$, which improve wear resistance. The C and Cu improve the thermal conductivity and thereby reduce the softening effect caused by temperature increases. Therefore, the abrasive particles of the composites are small and the wear marks are shallow.

3.  The SCFs randomly distributed in the matrix prevent the nucleation, deflection, and expansion of microcracks at the interface, release concentrated stress, and prevent material damage; therefore, the wear resistance of the composites is better than that of the Al6061 alloy.

Figure 10b shows that the phenomenon of delamination is significantly decreased, but some SCFs are debonded from the Al matrix. During wear testing, the surface temperature of the composite continues to increase, the material softens, and the interfacial bonding strength decreases. Under friction forces that exceed the interfacial bonding strength, the interface experiences debonding. The wear mechanisms of the Cu-SCF/Al composite are mainly fiber debonding and slight delamination wear. As is shown in Figure 10c, larger abrasive grains appear on the worn surface of the Ni-SCF/Al composite, with SCFs buried in the matrix. No fiber debonding occurs, but the hard intermetallic compound of Al$_3$Ni affects the interfacial quality. Each brittle intermetallic particle at the interface acts as a site for crack formation and subsequent propagation. During wear testing, the SCFs are fractured. The wear mechanisms are mainly fiber breakage accompanied by slight abrasion and adhesion. Figure 10d shows that smaller abrasive grains and shallower grooves appear on the Cu-Ni-SCF/Al worn surface than those occurring on the Cu-SCF/Al and Ni-SCF/Al composite surfaces. Because Cu-Ni-SCF/Al shows the highest interfacial bonding strength, matrix deformation is effectively prevented. In addition, the composite avoids the peeling off of large flakes and SCF debonding. The wear mechanism is slight abrasive wear. This indicates that the hardness of the Cu-Ni-SCF/Al composite is higher and its wear resistance is better, supporting the data reported in Sections 3.4 and 3.5. The occurrence of oxidative wear is indicated by O; Fe arises from component loss by the friction pair; C represents the ground SCFs; Ni and Cu represent the metal coating components on the SCF surfaces; and Al, Mg, and Si are alloy components.

Figure 11 shows the worn surfaces of the composites after the T6 heat treatment. Compared to the composite surfaces before T6 heat treatment, the worn surfaces are smoother with decreased delamination, further supporting the improvement of hardness and wear resistance after the T6 heat treatment of the composites.

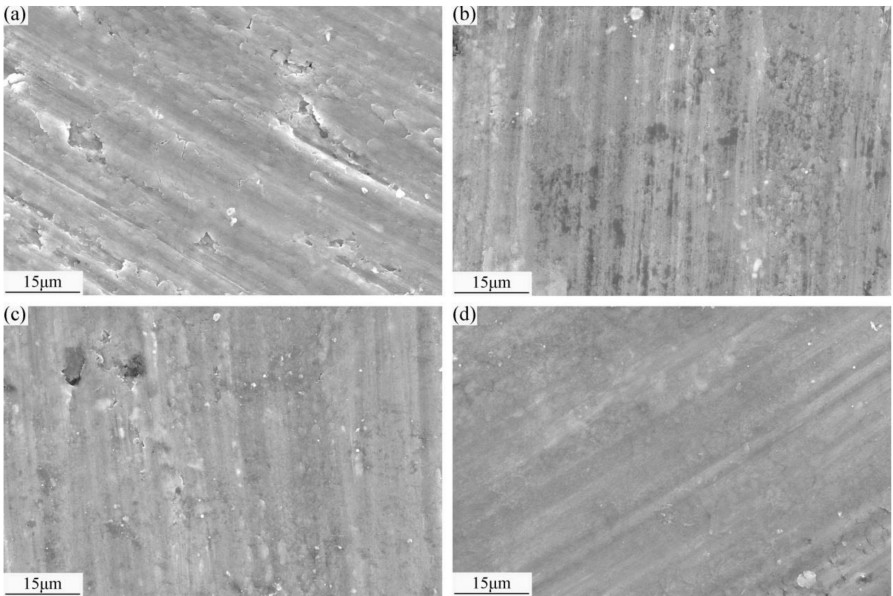

**Figure 11.** SEM images of the worn surfaces of the composites after the T6 heat treatment: (**a**) Al6061 alloy; (**b**) Cu-Cf/Al; (**c**) Ni-Cf/Al; and (**d**) Cu-Ni-Cf/Al.

### 3.6. Wear Debris

The SEM images and EDS patterns of wear debris are shown in Figure 12. Microcracks form on the worn surfaces of the materials under periodically variable friction and thermal stress, and crack

propagation causes the worn surface to break, flake, and form wear debris. The wear debris of the Al6061 alloy is large with flake-like shapes. The hardness of the Al6061 alloy is obviously lower than that of the AISI 52100 steel ball. During wear testing, the surface of the steel ball is in direct contact with the Al6061 alloy. The surface of the Al6061 alloy produces significant abrasive dust during plastic deformation. The EDS data shows O and Fe in the grinding dust. Fe comes from the e-like shapes. The hardness of the Al6061 alloy is obviously lower than that of the AISI 52100 steel ball, indicating that wear debris is produced by friction between the steel ball and the Al6061 alloy. The symbol O indicates the occurrence of oxidative wear. The wear debris of the SCF/Al composite is smaller and regular in size with particle-like shapes; under friction, the SCFs are squeezed and ground into graphite particles, which provide lubrication. Additionally, the hard metal coating on the SCF surface is ground to form an MML comprising a fine mixture of metallic oxides and C fragments at the interface of the friction pair, thus promoting the composites wear resistance. For this reason, the wear debris of the composites is smaller than that of the Al6061 alloy. The symbol C represents the ground fibers, Ni and Cu are the metal coating components on the fibers, and Al, Mg, and Si are alloy components. The Cu-Ni-SCF/Al composite yields the least abrasive debris, which is consistent with the previous analysis of the wear rates of the composite materials. The wear process is stable because small-sized wear debris has little influence on the worn surface. Large-sized wear debris deteriorates the surface and aggravates cutting and peeling wear; the wear quality loss is increased with increasing wear time. Therefore, the Cu-Ni-SCF/Al composite has the best abrasion resistance and the least quality loss.

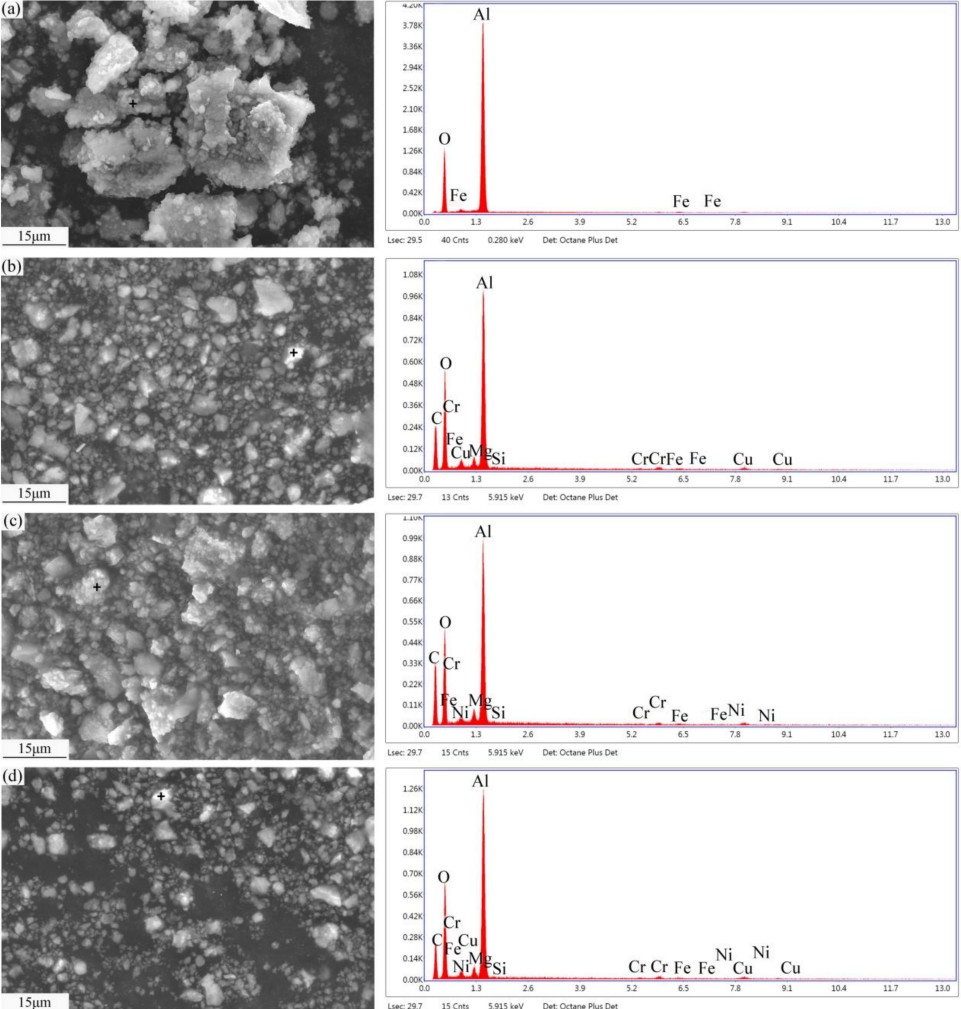

**Figure 12.** SEM images and EDS patterns of wear debris: (**a**) Al6061 alloy; (**b**) Cu-Cf/Al; (**c**) Ni-Cf/Al; and (**d**) Cu-Ni-Cf/Al.

## 4. Conclusions

In the present work, the addition of carbon fibers with coating improves the hardness, tensile strength and wear resistance of aluminum alloy. When the coated SCF content is 6 vol%, compared to the Ni-SCF/Al and Cu-SCF/Al composites, the Cu-Ni-SCF/Al composite shows the highest hardness, the lowest friction coefficient and wear rate, and the best load-carrying capacity and wear resistance. After heat treatment, the Cu-Ni-SCF/Al composites have the strongest interface binding strength. When the coated SCF content is 6 vol%, the hardness values of the Cu-Ni-SCF/Al composites are enhanced by 23.3% and the wear rate values of the Cu-Ni-SCF/Al composites are reduced by 49.04%, compared with that of the Al6061 alloy. The hardness, tensile strength and wear resistance are related to carbon fiber, coating, adhesion between substrate and lubrication of carbon fibers.

**Author Contributions:** S.D.: Conceptualization, data curation, writing—original draft, validation. H.Y.: Formal analysis, software. M.Y.: Funding acquisition, writing—review and editing. X.C. (Xiaochao Cheng): Writing—review and editing, resources. Y.F.: Funding acquisition, supervision, conceptualization. X.Z.: Supervision, writing—review and editing, resources. X.C. (Xiaoming Cao): Funding acquisition, conceptualization, supervision. All authors have read and agreed to the published version of the manuscript.

**Funding:** This work was supported by the Hebei Province Science and Technology Support Program (grant number 19274009D), the National Natural Science Foundation of China (grant number 51601056), and the Natural Science Foundation of Hebei Province of China (grant number E2017202012).

**Conflicts of Interest:** The authors declare that they have no known competing financial interests or personal relationships that could have appeared to influence the work reported in this paper. The funders had role in the design of the study. The authors have no conflict of interest.

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
