# Peer review of "Improved Friction and Wear Properties of Al6061-Matrix Composites Reinforced by Cu-Ni Double-Layer-Coated Carbon Fibers"

_metals, doi:10.3390/met10111542_

Round 1

Reviewer 1 Report

  1. I cannot find the Tables in the manuscript.

  1. Please change “GCr15-type 95 bearing steel” to a universal term (ex. “AISI 52100”).

  1. Fig. 1, 5: The color of lines cannot be recognized in the caption.

- Moreover, the thickness of coating should be presented.

  1. In section 2.5, what were the setting parameters (e.g., voltage, current, etc.) for XRD and SEM and EDS analyses.

- It is important because the diameter of fiber is approximately 10 um and the coating thickness seems to be less than 1 um. Those are very thin. The parameters will determine the penetration depth.

- In Fig.1 and Fig. 2, the authors assumed that the characterization conducted on the surface. However, if the coating will significantly thin, the characterization can be conducted not only the surface.

  1. Please add the explanations for the each figure in the caption.

- For example, Figure 1. SEM images of the interface microstructures of the modified SCFs. è Figure 1. SEM images of the interface microstructures of the modified SCFs for the (a) XXX, (b) XXX

- It will improve the readability. The present manuscript is extremely hard to read because it is difficult the match the figures and contents.

  1. In the experimental details, please add how the hardness was measured.

- How many times? Holding time? Load?

  1. Line 192: what kind of defects?

- Please mark the defects in SEM images (if it possible).

- At least, please add the references.

Author Response

Response to Reviewer

Dear Editor:

We would like to thank reviewers’ comments concerning our manuscript. We have studied the comments carefully and have made significant corrections which we hope meet with approval. Revised portion are marked in red in our revised manuscript. We would like to resubmit the manuscript entitled “Improved friction and wear properties of Al6061-matrix composites reinforced by Cu-Ni double-layer-coated carbon fibers”, which we wish to be considered for publication in “Metals”.

Comment 1: I cannot find the Tables in the manuscript.

Response 1: Thanks a lot for the reviewer’s comment. We appreciate the reviewer’s comment here. The whole manuscript has been corrected and polished carefully. We added the table again at the end of the article.

Comment 2: Please change “GCr15-type 95 bearing steel” to a universal term (ex. “AISI 52100”).

Response 2: We have corrected it in the manuscript, the revised parts are marked in red in the manuscript.

Comment 3: Fig. 1, 5: The color of lines cannot be recognized in the caption. - Moreover, the thickness of coating should be presented.

Response 3: The color of the lines in the drawing has been modified to be recognizable, and the corrected drawing is put back in the manuscript. It is a little difficult to calibrate the coating thickness. In the subsequent sintering process, there will be diffusion between the fiber, coating and substrate, and there will be no obvious boundary between coating and substrate.

Comment 4: In section 2.5, what were the setting parameters (e.g., voltage, current, etc.) for XRD and SEM and EDS analyses.

- It is important because the diameter of fiber is approximately 10 um and the coating thickness seems to be less than 1 um. Those are very thin. The parameters will determine the penetration depth.

- In Fig.1 and Fig. 2, the authors assumed that the characterization conducted on the surface. However, if the coating will significantly thin, the characterization can be conducted not only the surface.

Response 4: The specific parameters have been added in the revised manuscript, where EDS signals do not penetrate the fibrous part of the coating. There is some carbon in the EDS spectrum as shown in Figure 2 (b) and (f), because there is a diffusion between C and Ni after the fibre has been treated, rather than a signal excited by the electron beam passing through the coating.

In the XRD pattern, the x-rays penetrate the coating because they are more penetrating, and there are some peaks in the diffraction pattern of the carbon fiber.

Comment 5: Please add the explanations for the each figure in the caption.

- For example, Figure 1. SEM images of the interface microstructures of the modified SCFs. è Figure 1. SEM images of the interface microstructures of the modified SCFs for the (a) XXX, (b) XXX

- It will improve the readability. The present manuscript is extremely hard to read because it is difficult the match the figures and contents.

Response 5: We have added a detailed explanation for each drawing and marked it in red in the manuscript.

Comment 6: In the experimental details, please add how the hardness was measured.

- How many times? Holding time? Load?

Response 6: The experimental details of the hardness test are as follows and added to the manuscript. The hardness of the material reflects the performance of the sintered sample. This experiment used Shimadu HMV-2t Vickers microhardness tester (HV0.1=100MPa, 10s) to test the hardness of the composite, ten points were selected for each sample to be tested and averaged.

Comment 7: Line 192: what kind of defects?

- Please mark the defects in SEM images (if it possible).

- At least, please add the references.

Response 7: We supplemented the experiment and added a figure (Fig.7) to the manuscript, the defect can be clearly seen in Figure 7.

Reviewer 2 Report

The subject of the manuscript is very interesting. However, my impression is that it was prepared hastily and is only a research report. There are no tables and subscripts in the file that I downloaded (as indicated in the attached file). Figures 5 and 6 do not describe the individual volume fraction of the reinforcement or the coating deposited on the fibers. There was no indication of how the proportion of the reinforcement was determined, or why such and not other fiber volume fractions were selected for the production of composites. In my opinion, the article requires improvement both in terms of the method of producing composites and the final summary of the results.

Author Response

The response to the reviewer’s comments

Manuscript ID metals-972437

Title: Improved friction and wear properties of Al6061-matrix composites reinforced by Cu-Ni double-layer-coated carbon fibers

Shuhan Dong 1, Huiyong Yuan 2, Xiaochao Cheng 1, Yongzhe Fan 1, Mingxu Yang 1,3,, Xue Zhao 1*, and Xiaoming Cao 1

1 School of Materials Science and Engineering, Hebei University of Technology, 29 Guangrong Road, Tianjin 300132, PR China

2 Bei jing Sunwise Intelligent Technology Co., Ltd. 16 Nansan Street, Zhongguancun, Beijing 100080, PR China

3 Beijing Jitai cold-forging technology Co., Ltd. 9 Zhongguancun South Avenue, Beijing 100081, PR China

*Correspondence: zhaoxue@hebut.edu.cn (Z. X.),  mingxu_yang@hotmail.com (Y. M. X.)

Response to Reviewer

Dear Reviewer:

We would like to thank reviewers’ comments concerning our manuscript. We have studied the comments carefully and have made significant corrections which we hope meet with approval. Revised portion are marked in red in our revised manuscript. We would like to resubmit the manuscript entitled “Improved friction and wear properties of Al6061-matrix composites reinforced by Cu-Ni double-layer-coated carbon fibers”, which we wish to be considered for publication in “Metals”.

Comment 1: There are no tables and subscripts in the file that I downloaded (as indicated in the attached file).

Response 1: We added all the tables at the end of the revised manuscript. The subscripts has also been added and marked in red in the revised manuscript.

Comment 2: Figures 5 and 6 do not describe the individual volume fraction of the reinforcement or the coating deposited on the fibers.

Response 2: We added a specific description for Figure 5 and 6. The additions are highlighted in red in the manuscript.

Comment 3: There was no indication of how the proportion of the reinforcement was determined, or why such and not other fiber volume fractions were selected for the production of composites.

Response 3: Before determining the experimental parameters, we conducted a large number of control variable experiments and selected other parameters, and finally determined the parameters written into the manuscript. In order to keep the paper concise and not redundant, we chose the most representative data in figure 8 and Figure 9 of the manuscript. It can be seen from Figure 8 and Figure 9 in the manuscript that when the fiber content is 6%, the hardness is the highest and the wear rate is the lowest. It can be seen from figure 1, 2 and 3 below that when the fiber content is 6%, the linear expansion coefficient of the composite material is the smallest, the wear rate is the smallest, and the tensile strength is the largest. FIG. 4 shows that the friction coefficient of the composite material is not minimum when the fiber content is 6%. But the composite material has the best comprehensive performance when its fiber content is 6%.

Figure 1. The linear expansion coefficient of different composites before and after heat treatment (a-Ni-Cf/Al, b-Cu-Cf/Al, c-Cu-Ni-Cf/Al)

Figure 2. The wear rate of different composites before and after heat treatment

(a-Ni-Cf/Al, b-Cu-Cf/Al, c-Cu-Ni-Cf/Al)

Figure3. The tensile strength of different composites before and after heat treatment

(a-Ni-Cf/Al, b-Cu-Cf/Al, c-Cu-Ni-Cf/Al)

Figure 4. Diagram of friction coefficient and fiber content of different composites

Comment 4: In my opinion, the article requires improvement both in terms of the method of producing composites and the final summary of the results.

Response 4: The specific parameters of the composite fabrication process are described in detail in tables, all of which are attached at the end of the manuscript. The final summary of the results department made further adjustments and marked red in the revised manuscript.

Comment 5: Error on line 32.

Response 5: The error on line 32 is an editing error and has been deleted from the revised manuscript.

Comment 6: Interpretation of lines 56-58 of the original manuscript and lines 56-58 of the present manuscript.

Response 6: When carbon fiber plating copper, if the process control is not good, copper coating and carbon fiber between a part of the defects, such as pores and destroy the coating and fiber bond. Compared with the carbon fiber obtained by the intact coating process, the bonding force between the carbon fiber and the coating is smaller, and this kind of bonding mode becomes the low-strength mechanical bonding.

Adding carbon fiber to composites in this way can cause some problems, such as the fiber desquamation described in this article. It will also reduce the density of the composite material, and thus affect the hardness, tensile strength and wear resistance of the material.

Round 2

Reviewer 1 Report

Please check the minor typos in the manuscript.

Ex) Line 104: Scanning power 2kw --> scanning power 2 kW

Reviewer 2 Report

Because I am not an expert, but I suggest language proofreading by a native speaker. In terms of content  I believe that the manuscript may be published in the Metals journal in its current form.